# The Influence of Wet Feed pH on the Growth of *Tenebrio molitor* Larvae

**Carl L. Coudron ***, **David Deruytter** and **Jonas Claeys**

Inagro, Ieperseweg 87, 8800 Roeselare, Belgium; david.deruytter@inagro.be (D.D.); jonas.claeys@inagro.be (J.C.)
* Correspondence: carl.coudron@inagro.be

**Abstract:** For optimal growth, *Tenebrio molitor* needs both dry feed and wet feed. Storing dry feed is not a problem, but storing wet feed over a prolonged period is more challenging due to spoilage. It could be stored in a refrigerated room, but this process is energy consuming and therefore increases the price of production. Another option is to ferment the feed, as is done regularly in other branches of agriculture. No energy is needed, and the feed remains stable due to low pH levels. In this study, we assessed the growth of mealworm larvae fed with wheat bran and agar-agar gel. Different treatments received agar-agar gel of a specific pH, varying between 3 and 9 in increments of one pH unit, resulting in seven assessed pH values. The average weight of the larvae was determined every week until maximum weight was achieved. Mealworms at harvest grown at the lowest pH (3.02) were on average 8.1% lighter than their counterparts grown at higher pH levels. However, within ranges that could realistically occur in a mealworm production setting (pH > 3.5), no significant differences were found. In conclusion, fermentation can be used to store mealworm wet feed, without pH having a detrimental effect on mealworm growth.

**Keywords:** mealworm; yellow mealworm; moisture source





## 1. Introduction

Worldwide, the mealworm (*Tenebrio molitor*) is one of the most commonly reared insect species as a source of protein for food and feed applications. Over the last decades, a large number of studies have been published on how to rear this insect, initially as a model organism for studies related to storage pests and later as (potentially) an alternative protein source [1–5].

A conventional way of rearing mealworms involves the use of a cereal-based dry feed (e.g., wheat bran) that serves both as feed and as bedding material. This dry feed is supplemented with a wet feed that is rationed on a regular basis and predominantly serves as a source of water for the mealworms. Fresh vegetables (such as carrots) or residual streams from vegetable production (such as cucumber foliage) are suitable candidates to serve as a wet feed. However, for optimal usage it is preferred to break the epidermis of these vegetables (by cutting or shredding) to make the water more accessible to the mealworms and reduce rationing portions. This requires the preparation of fresh shredded vegetables prior to each (wet) feeding on a regular basis. One way of dealing with this inconvenience, bypassing the need for energy-inefficient cool storage, is to use anaerobic fermentation to preserve shredded vegetables for extended periods of time. Moreover, this could be a way to prolong the availability of temporarily available residual streams. During anaerobic fermentation, bacteria and eukaryotes metabolise carbohydrates into organic acids or alcohols. These organic acids cause a drop in pH to a point where most microbial growth is inhibited.

Feed pH has been shown to affect feeding behaviour in *Rhagoletis pomonella* (Diptera) [6] and larval development in *Ceratitis capitata* (Diptera) [7] and *Hermetia illucens* (Diptera) [8,9]. Moreover, pH in the digestive tract plays an important role in the activity of digestive

enzymes, and insects have been shown to actively regulate the pH in their digestive systems [10]. The digestive system of mealworms is composed of different regions and each has a specific pH range. Terra et al. [11] observed pH values of 6.0, 5.6 and 7.9 in the foregut, anterior midgut and posterior midgut, respectively; each region exhibited differences in enzyme abundance as well. Offering acidic or basic food to mealworms may affect their feeding behaviour or cause increased energy requirements to regulate the pH in their digestive tract, and in the worst case cause a shift in the activity of digestive enzymes and therefore affect the intake of nutrients.

The goal of this study was to assess the influence of the pH of wet feed on the growth and development of mealworm larvae in order to determine whether anaerobic fermented—and therefore storable—wet feed can be used in *Tenebrio molitor* rearing systems.

## 2. Materials and Methods

### 2.1. Mealworm Colony

The mealworm colony used in this study has been kept at the Inagro Insect Research Centre since 2013. They are kept in 60-by-40 cm plastic crates (with an inner surface area of 2000 cm$^2$) at a temperature of 27 $\pm$ 0.3 °C, 60 $\pm$ 2.5% relative humidity and in the dark, except during feeding. The animals are fed ad libitum with INSECTUS Mealworm Grow (Mijten nv, Belgium) and chopped and fermented chicory roots. The CO2 concentration is monitored and kept below 1500 ppm by ventilation.

### 2.2. Experimental Setup

Agar-agar gel was used as a wet feed in this experiment as it has no nutritional value but is an excellent source of water for the mealworms [12]. The pH level of the agar-agar gel was modified by adding an inorganic acid or base. Inorganic acids and bases were preferred over organic ones as they have no caloric value. Agar-agar gels with seven different pH levels were used (3.02, 3.98, 4.90, 6.04, 6.98, 8.01 and 9.01). To decrease the pH of the agar-agar gel, $H_3PO_4$ was added to demineralised water; KOH was used to increase the pH; and for the neutral pH, pure demineralised water was used. The demineralised water was stirred continuously while $H_3PO_4$ or KOH was added to the solution, and the pH was monitored (Memosens pH-elektrode FL S 93-225 MF NMSN, VWR, Leuven, Belgium) until the desired level was reached. Agar-agar powder was added to each solution (25 g/L, Brouwland nv, Beverlo, Belgium), and all the solutions were then brought to boiling point. After boiling and before solidification, pH values were again checked. The boiled liquids were then poured out to cool and solidify. The solidified gel was then cut into cubes of one-by-one cm. The agar-agar cubes were stored in a refrigerator to preserve them for the entirety of the experiment. It was assumed that the gelling did not affect pH values.

The mealworms that were used for the experiment were produced as follows. Beetles (eight crates containing 250 g of beetles each) were allowed to oviposit for one week in INSECTUS Mealworm Grow. After this oviposition week, the beetles were separated from the substrate (and their eggs) by sieving with a vibrating screen (2 mm), and the eggs were left to hatch and grow undisturbed for three weeks. After these three weeks, the contents of all 8 crates were merged, thoroughly homogenized and three representative samples were collected. The mealworms in each sample were counted and mean mealworm density (number of mealworms per gram of substrate) was estimated. This estimate was used to establish 21 portions of mealworm–substrate mixture with an estimated 5000 mealworms each. By this point, the mealworms had grown to a size of, on average, 1.7 mg (weighed on a ME203T, Mettler Toledo). Each batch of 5000 mealworms was assigned to a rearing crate and supplied with wheat bran ad libitum. From then on, the mealworms were provided with portions of agar-agar gel of a specific pH level. Every day the crates were checked for the presence of agar-agar and, if needed, fresh agar-agar was added. In total, 7 treatments were tested in triplicate.

### 2.3. Measurements

Mealworm growth was monitored by determining the average weight of the mealworms one, two, three and four weeks after the start of the experiment. The content of each crate was gently homogenised and a sample of the substrate was taken. All mealworms were isolated from the sample, counted and weighed (ME203T, Mettler Toledo), and after weighing the mealworms were put back in their respective crates. If fewer than 100 mealworms were present in a sample, a new sample was taken in order to obtain an accurate estimate of individual weight. After 5 weeks, the experiment was terminated, as pupae started to appear in the crates, indicating that the mealworms were near their maximal weight. Each crate was harvested by separating the frass from the larvae with a vibrating screen (mesh size of 2 mm). The total live mealworm yield was determined as well as the average individual mealworm weight.

### 2.4. Statistical Analysis

A growth model was constructed using R statistical software. A linear mixed-effect model was used (nlme package) to assess the influence of time (continuous) and pH (treated as a categorical variable with 6 levels; pH 3.02 was used as a reference level) on the growth of mealworm larvae. The mixed-effect modeling was necessary due to the longitudinal nature of the data. Both replicates and time were added as random effects. Mealworm growth was expressed as a cubic equation of time (M). The cubic term was necessary to deal with the stagnation of mealworm growth near the end of the experiment. An interaction between time and pH was added in order to fit the potential pH-dependent slope of the growth curve. Predictors were assessed for significance based on *p*-value (<0.05).

$$\text{Log10 (MW)} = T + T^2 + T^3 + pH + T \times pH \text{ (M)}$$

where MW = 'mean mealworm weight' (mg), T = time (weeks) and pH = pH level of the wet feed.

Finally, to evaluate the harvest parameters (total live yield, average mealworm weight and number of mealworms per crate) an ANOVA with a post hoc Tukey test was applied, or, when ANOVA assumptions were violated, a Wilcoxon rank-sum test was performed.

## 3. Results

The results of modelling mealworm growth are presented in Table 1 and visualized in Figure 1. The pH level as a stand-alone term did not significantly differ between levels, indicating that no differences between intercept values for the different treatments were found. However, the interaction with time did result in some significant differences from the reference level (pH of 3.02). pH 4.9, 6.04, 8.01 and 9.01 had significantly higher slopes. On average, the difference in weight increased around 2.1% every week, while, for the non-significantly different pH levels, 3.98 and 6.98, the increase in weight difference was only 1.39% and 1.76%, respectively, every week.

Harvest parameters are shown in Figure 2. On average, 5093 mealworms (±203 SD) were present in each crate and no significant differences were found between treatments. Live mealworm biomass yield per crate was, on average, 649 g (±25.3 SD) and did not differ significantly between treatments. Only for the average mealworm weight at harvest was a significant difference found and only between the pH levels 3.02 and 6.98. Mealworms that were grown in the lowest pH conditions were on average 7.4 mg lighter than these grown at a neutral pH. No other significant differences were found.

**Table 1.** The mealworm larvae growth model. T = time (weeks), DF = degrees of freedom, SE = standard error.

|  | Estimate | SE | DF | t-Value | *p*-Value |
|---|---|---|---|---|---|
| Intercept | 0.232 | 0.00965 | 108 | 24.1 | 0.00 * |
| $T^3$ | −0.0128 | 0.000584 | 108 | −21.9 | 0.00 * |
| $T^2$ | 0.0667 | 0.00445 | 108 | 15.0 | 0.00 * |
| T | 0.354 | 0.00936 | 108 | 37.9 | 0.00 * |
| pH (3.98) | 0.00826 | 0.0127 | 108 | 0.649 | 0.52 |
| pH (4.90) | −0.0120 | 0.0127 | 108 | −0.946 | 0.35 |
| pH (6.04) | −0.00116 | 0.0127 | 108 | −0.091 | 0.93 |
| pH (6.98) | −0.00229 | 0.0127 | 108 | −0.180 | 0.86 |
| pH (8.01) | −0.00664 | 0.0127 | 108 | −0.522 | 0.60 |
| pH (9.01) | −0.0136 | 0.0127 | 108 | −1.07 | 0.29 |
| T × pH (3.98) | 0.00599 | 0.00420 | 108 | 1.42 | 0.16 |
| T × pH (4.90) | 0.00924 | 0.00420 | 108 | 2.20 | 0.03 * |
| T × pH (6.04) | 0.00869 | 0.00420 | 108 | 2.07 | 0.04 * |
| T × pH (6.98) | 0.00759 | 0.00420 | 108 | 1.80 | 0.07 |
| T × pH (8.01) | 0.00867 | 0.00420 | 108 | 2.06 | 0.04 * |
| T × pH (9.01) | 0.00914 | 0.00420 | 108 | 2.17 | 0.03 * |

* Indicates a significant predictor (*p*-value < 0.05).

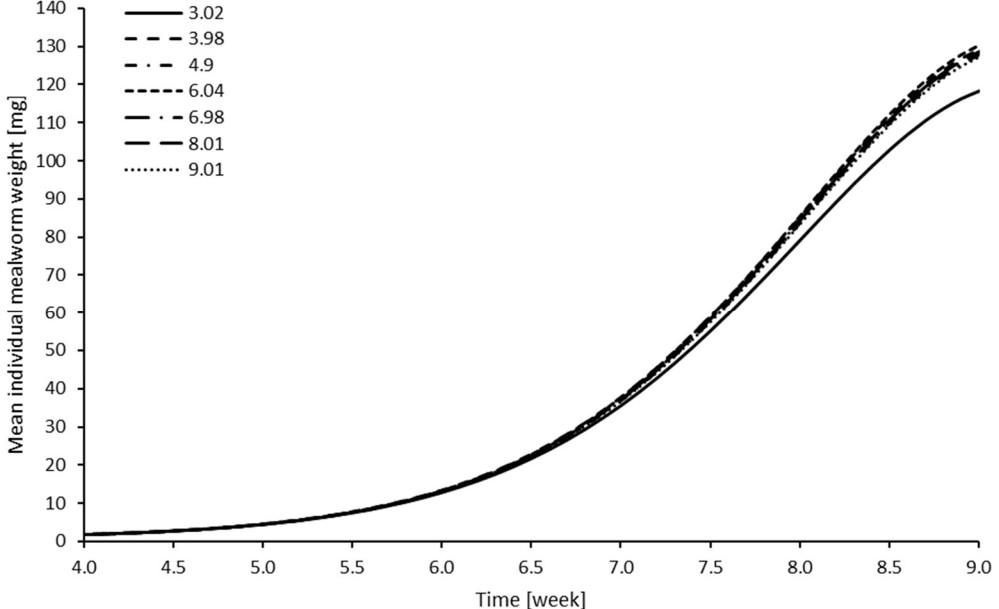

**Figure 1.** Visualisation of the growth model (M), with average mealworm weight (mg) expressed over time (weeks) for the different pH values of the wet feed. The start of oviposition was chosen as point zero; the experiment started at week four.

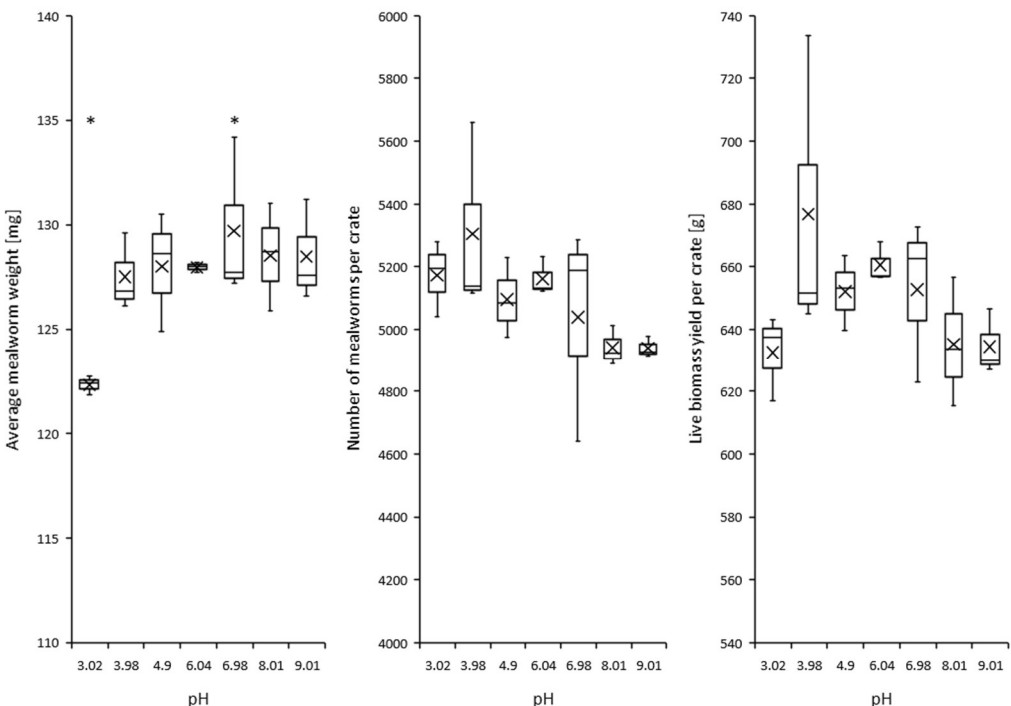

**Figure 2.** Boxplot representation of harvest parameters. Asterisks (*) indicate significant differences between different pH values (*p*-value < 0.05).

## 4. Discussion

Despite the fact that significant amounts of wet feed are consumed by mealworms during their growth (at least 135% of their final live body weight) [13], a wet feed pH as low as 3.98 does not seem to affect their growth. At the lowest tested pH of 3.02 there were some indications of reduced growth. However, the differences were rather subtle. When only harvest parameters were compared, no differences could be differentiated, except for a lower average mealworm weight with a pH of 6.98. With the growth model, more significant differences could be observed but only with four of the six other treatments. Based on the growth model, mealworms at harvest grown at the lowest pH were on average 8.1% lighter than their counterparts.

To the authors' knowledge, no other studies have been published on the influence of (wet) feed pH on the growth of mealworms. For other insect species reared for feed applications, such as the black soldier fly (*Hermetia illucens*), published data are available [8,9]. Pang et al. [8] observed virtually no growth of larvae in a substrate at pH 3. However, it should be noted that rearing conditions for the two species are completely different, as black soldier fly larvae live in their feed and are therefore directly exposed to the pH of their surroundings, meaning that other factors apart from digestion might come into play. As for mealworms, they live in a dry substrate and are only exposed to the pH of their wet feed while consuming it.

The pH that can be encountered after natural fermentation of vegetables or their side streams will vary depending on several factors, such as active microbial species or the nature of the fermented substrate. A lower limit after lactic acid fermentation seems to be around 3.5, as below this value microbial growth is inhibited. Thus, it seems unlikely that pH values as low as 3 will be encountered in a mealworm-production environment where naturally fermented feeds are used. Mealworm producers can therefore be reassured that the pH of wet feed will not negatively affect the growth and development of their mealworms. Prepared wet feed can be stored cheaply for prolonged periods under anaerobic conditions, not only avoiding the need for cooled storage but saving time as well, as a stock can be built up for several weeks. This bypasses the need for daily feed preparation. Moreover, sometimes wet feed ingredients are not available all year round; fermentation is

thus a safe way to overbridge these periods of shortage. In such a scenario, fermentation might be favourable over switching wet feeds, as this might affect the composition of the end product. Thus, fermentation of insect feed can be one of the ways of guaranteeing a more stable and predictable production process and larval composition.

However, other factors might affect the quality of wet feed as a consequence of fermentation, such as losses of easily accessible carbohydrates during an initial aerobic phase of anaerobic fermentation or the transformation of carbohydrates (e.g., glucose) into less caloric organic molecules (e.g., lactate). Another concern could be possible increase in the bioavailability of soilborne heavy metals under acidic conditions [14] caused by fermentation, especially when uncleaned below-ground plant parts, such as bulbs, tubers or taproots, are used as wet feed material. Increased bioavailability of heavy metals could then lead to possible accumulation in the mealworms [15]. In addition, the influence of extremely acidic feed on the physiological response of mealworm intestines presents an interesting topic for further research. These factors should be assessed in future experiments.

In conclusion, the pH of wet feed does not seem to affect the growth of mealworm larvae within ranges that would be encountered in a production setting, e.g., the low pH values that arise during natural fermentation of wet feed.

**Author Contributions:** C.L.C. was involved in data acquisition, statistical analysis and original manuscript draft preparation. D.D. was involved in conceptualization and methodology and reviewing of the manuscript draft. J.C. was involved in reviewing the manuscript draft and funding acquisition. All authors have read and agreed to the published version of the manuscript.

**Funding:** This research has received funding from the European Union's Horizon 2020 research and innovation program under grant agreement No. 861976.

**Institutional Review Board Statement:** Not applicable.

**Informed Consent Statement:** Not applicable.

**Data Availability Statement:** Not applicable.

**Acknowledgments:** The authors would like to thank L. De Praetere and A. Devos for the many larvae counts.

**Conflicts of Interest:** The authors declare no conflict of interest.

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
