# Peer review of "The Influence of Wet Feed pH on the Growth of Tenebrio molitor Larvae"

_sustainability, doi:10.3390/su14137841_

Round 1

Reviewer 1 Report

The study submitted to the journal is straightforward and easy to follow. Even though it seems to be rather kind of "short communication" than "full research paper", I like it and can hardly find anything to criticize. 

In the introduction, some relevant sources dealing with mealworm rearing can be added to have at least some insight into "a lot of studies".

In M&Ms, I wonder whether breeding is the correct term for what you are doing with the mealworms. According to the literature, the verb "to breed" means to keep animals with a focus on improving their performance. Therefore, I think that "to rear" is the more correct verb from this point of view.

Regarding results, maybe the final weights (after 5 weeks) can be added to the table. 

In L168, the citation may read as just "Pang et al."

In the references, authors should unify the format of journal names (sometimes they capitalize, sometimes not).

Thank you for the opportunity to review this research.

Author Response

The authors want to thank the reviewer for their time to read and comment on the manuscript. The reviewers comments were answered below by the authors in italics.

In the introduction, some relevant sources dealing with mealworm rearing can be added to have at least some insight into "a lot of studies".

A selection of references over different decades was added.

In M&Ms, I wonder whether breeding is the correct term for what you are doing with the mealworms. According to the literature, the verb "to breed" means to keep animals with a focus on improving their performance. Therefore, I think that "to rear" is the more correct verb from this point of view.

The authors agree and changed the wording in this phrase. (L59)

Regarding results, maybe the final weights (after 5 weeks) can be added to the table.

The authors chose not to mention the data in the table as it is specifically meant to display the growth curves. Moreover, the data is already visualised in the boxplot.

In L168, the citation may read as just "Pang et al."

This was adjusted.

In the references, authors should unify the format of journal names (sometimes they capitalize, sometimes not).

The authors homogenised the references.

Reviewer 2 Report

This work investigated the effect of different pH values on the growth of T. molitor larvae. The experiments consisted of offering agar-agar gels with pH values between 3 and 9 as a source of moisture and quantifying larvae development parameters until they reached their maximum size. Only at the lowest pH there was impairment of larval growth. The authors conclude that the pH values observed in fermented vegetables should not negatively affect the larvae. This observation is obvious, as producers already use this means of preserving wet feeds. A major contribution would be the investigation of what happens in the intestines of larvae when they ingest very acidic diets.

Overall, the paper is slim, but well written and clear. The experimental set-up is sound, and the data give some new useful information. 

Here is a suggestion the authors should address to improve their manuscript:

1-    The authors could have measured the pH of the agar-agar gels with contact electrodes to be sure of the values that the larvae actually ingested in the crates.

2-    In line 193 the authors must consider that lactate is not a carbohydrate.

Author Response

The authors want to thank the reviewer for their time to read and comment on the manuscript. The reviewers comments were answered below by the authors in italics.

The authors conclude that the pH values observed in fermented vegetables should not negatively affect the larvae. This observation is obvious, as producers already use this means of preserving wet feeds. A major contribution would be the investigation of what happens in the intestines of larvae when they ingest very acidic diets.

The authors were not aware of the fact using fermented feeds in mealworm rearing was common practice. Common practice in the Benelux is using fresh carrots and preparing these on a daily basis. The authors agree that the physiological response of the mealworms’ intestines would be valuable information. A line concerning this topic was added in discussion (L 198-200).

The authors could have measured the pH of the agar-agar gels with contact electrodes to be sure of the values that the larvae actually ingested in the crates.

The authors clarified this issue (L76-77; 80).

In line 193 the authors must consider that lactate is not a carbohydrate.

The usage of the word carbohydrate was omitted and replaced with “organic molecules”.